



# Vertical concentrations gradients and transport of airborne microplastics in wind tunnel experiments.

Eike Maximilian Esders[1,*], Christoph Georgi[2], Wolfgang Babel[1,3], Andreas Held[2], and Christoph Karl Thomas[1,3]

[1]Micrometeorology Group, University of Bayreuth, Universitätsstraße 30, Bayreuth, Germany
[2]Institute of Environmental Technology, Technische Universität Berlin, Straße des 17. Juni 135, Berlin, Germany
[3]Bayreuth Center of Ecology and Environmental Research, Dr.Hans-Frisch-Str.1-3, Bayreuth, Germany
[*]**Correspondence**: Eike Esders, eike.esders@uni-bayreuth.de

**Abstract.** Microplastics are an ubiquitous man-made material in the environment, including the atmosphere. Little work focused on the atmospheric transport mechanisms of microplastics and its dispersion, despite being a potential pollutant. We study the vertical transport of airborne microplastics in a wind tunnel as a controllable environment with neutral stability, to identify the necessary conditions for long-range atmospheric transport of microplastics. An ultrasonic disperser generated
airborne water droplets from a suspension of polystyrene microspheres (MP) with a diameter of 0.51 $\mu$m. The water droplets were injected into the airflow, evaporating and releasing single airborne MPs. The disperser allowed for time-invariant and user-controlled concentrations of MP in the wind tunnel. MP were injected at 27, 57, and 255 mm above ground. A single GRIMM R11 optical particle counter (OPC) and three Alphasense OPCs measured time-averaged MP concentration profiles (27, 57, and 157 mm, above ground). These were combined with turbulent airflow characteristics measured by a hot-wire
probe to estimate vertical particle fluxes using the flux-gradient similarity theory. The GRIMM R11 OPC measured vertical concentration profiles by moving its sampling tube vertically. The three Alphasense OPCs measured particle concentrations simultaneously at three distinct heights. Results show that maximum concentrations are not measured at the injection height, but are shifted to the surface by gravitational settling. The MP experience higher gravitational settling while they are part of the larger water droplets. For the lowest injection at 27 mm, the settling leads to smaller MP concentrations in the wind tunnel,
as MPs are lost to deposition. Increasing the wind speed decreases the loss of MP by settling, but settling is present until our maximum friction velocity of 0.14 ms$^{-1}$. For the highest injection at 255 mm and laminar flow, the settling resulted in a net MP emission, challenging the expectation of a net MP deposition for high injection. Turbulent flows reverse the MP concentration profile giving a net MP deposition with deposition velocities of 3.7 ± 1.9 cm s$^{-1}$. Recognizing that microplastics share deposition velocities with mineral particles bridges the gap in understanding their environmental behavior. The result supports the
use of existing models to evaluate the transport of microplastics in the accumulation mode. The similar deposition velocities imply, that atmospheric transported microplastics can be found in the same places as mineral particles.



## 1 Introduction

The introduction of plastic products started in the 1950s and had a great impact on the medical, industrial, agricultural sectors (Ostle et al., 2019). Yearly production of plastic was 280 million metric tons in 2012 and is expected to rise to 33 billion metric tons in 2050 (Rochman et al., 2013). Commonly, microplastics (MP) are defined as plastic particles with a diameter of less than 5 mm. Studies indicated that residual plastic particles can be found in every environmental compartment on our planet (Allen et al., 2022), including remote locations such as national parks (Brahney et al., 2020) and the Arctic (Bergmann et al., 2022). Especially for remote locations, atmospheric transport is a main pathway (Evangeliou et al., 2020) that had been neglected until Dris et al. (2015) detected MP in deposition samples in greater Paris. Since then, more studies investigated the atmospheric deposition (Klein and Fischer, 2019; Kernchen et al., 2022) the emission to the environment (Chen et al., 2021) and the environmental MP life cycle inventories (Croxatto Vega et al., 2021) covering various aspects of long scale transport. On smaller scales, air-land interface processes of MP emission and deposition can be investigated in the field (Rezaei et al., 2019) or in the laboratory (Bullard et al., 2021; Esders et al., 2022) using a wind tunnel as an idealized and controllable environment. Bullard et al. (2021) pointed out the distinct behavior of mineral particles and MP while Rezaei et al. (2019) investigated the erosivity of low density MP. While MP properties are important, the characteristics of the wind flow and turbulence also play a major role for the vertical transport and therefore resuspension rates of MP into the air (Esders et al., 2022). This study focuses on small-scale vertical transport of MP. We introduce a time-invariant concentration of MP in a controlled wind tunnel environment and measure the vertical particle concentration gradients using multiple optical particle counters. Further, the flow conditions are varied to study its influence on vertical transport. We address the following research questions: How do vertical particle concentration profiles vary with flow conditions and particle injection heights? Are vertical particle fluxes derived from particle concentration gradients consistent with commonly applied parametrization of turbulent vertical particle transport?

As previous experiments in the wind tunnel gave statistically robust estimates for the suspension potential of larger particles (Esders et al., 2022). Now we are performing additional particle transport experiments, expecting that vertical particle transport induced by turbulence is consistent with theoretical deposition velocities. To our knowledge this is the first study to observe the vertical gradients and deposition velocities of airborne microplastics under controlled conditions.

## 2 Methods

### 2.1 Wind tunnel

The experiments were carried out in a wind tunnel with a total length of 7.3 m, 0.6 m width and 1.2 m height, and a contraction zone with a cross-section of 270 mm × 540 mm (width × height) (see Figure 1; Esders et al., 2022). Twelve fans (RAB O TURBO 250, DALAP GmbH, Germany) were operated at one end of the wind tunnel to adjust the airflow. Four different flow conditions with a mean horizontal wind speed ranging from 0.38 to 2.17 m $s^{-1}$ at a height of 157 mm above the wind tunnel surface were used in the experiments. A honeycomb structure ensured laminar flow conditions at the inflow section of the wind tunnel, followed by a defined pattern of cones and roughness elements that generated shear-driven turbulence. Turbulence



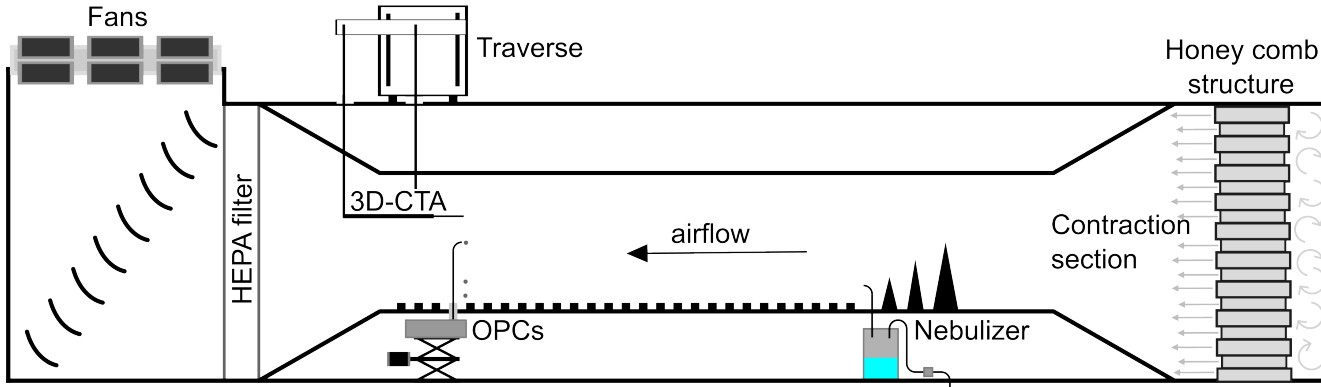

**Figure 1.** Concept of the wind tunnel with roughness elements, nebulizer, optical particle counters (OPCs) with three variable inlet heights and a three-dimensional constant temperature anemometer. The airflow is straightened before entering the wind tunnel by a honey comb structure.

measurements were performed using a 3D constant temperature anemometer (Model 55P95, controller, Model 54T42, Dantec Dynamics, Skovlunde, Denmark; hereafter referred as hotwire). The hotwire was mounted to a remotely controlled traverse at the end of the contraction zone. Horizontal mean wind speed ($U$) and friction velocity ($u_*$) were measured at six heights of 13, 16, 27, 57, 128 and 157 mm above the wind tunnel surface using the eddy covariance technique. The raw data was processed with the software package bmmflux (see appendix in Thomas et al., 2009). MP were introduced to the airflow in the inflow section of the contraction zone, and measured at the end of the contraction zone.

## 2.2 Aerosol generation

Factory fresh polystyrene microspheres (hereafter, referred as MP) with a nominal diameter of 0.51 $\mu$m (Polybead Microspheres, Polysciences, Hirschberg an der Bergstraße, Germany) were provided in a suspension (2.7 % mass) and further diluted to a concentration of $3.9 \times 10^9$ particles per mililiter. The nebulizer consisted of a glass reservoir filled with the MP suspension and a submerged 24 V ultrasonic nebulizing unit operating at about 1.7 MHz (Mist Fogger, FITNATE, PR China). Nebulizing the MP suspension at 1.0 ml min$^{-1}$, approximately 3.9 x $10^9$ airborne particles were generated per minute. After one hour, the glass reservoir was refilled with 60 ml of the MP suspension to maintain the water column within 1 cm of the optimum level for stable particle generation. Three air pumps with a flow rate of 1.2 l min$^{-1}$ connected in parallel, generated a flow rate of 3.6 l min$^{-1}$. The air pumps introduce filtered air to the reservoir and thereby advected airborne MP into the wind tunnel. The tubing of the nebulizer was made of copper with an inner diameter of 5 mm and conductive silicone tubing with an inner diameter of 6 mm, minimizing electrostatic particle losses. The generated aerosol was injected at three different heights above the wind tunnel surface, at 27 mm, 57 mm, and 255 mm. In the wind tunnel, the generated droplets quickly dried while traveling in the





airflow at a relative humidity < 25 %, yielding dry MP with a diameter of 0.51 $\mu$m. In a control experiment, the nebulizer was run with only demineralized water to obtain a baseline of particles generated from residual impurities in the water.

### 2.3    Particle measurement

Two types of optical particle counters (OPC) were used to detect airborne MP. A 32-channel optical particle spectrometer (Model R11, GRIMM Aerosoltechnik, Ainring, Germany) was attached to a remotely controlled laboratory jack to sample at different heights. The GRIMM R11 was operated with a sample flow rate of 1.2 l min$^{-1}$ and a measurement interval of 6 s. Additionally, a set of three low-cost OPCs (OPC-N3, Alphasense, Essex, United Kingdom) was installed at 27, 57 and 157 mm above the wind tunnel surface. The Alphasense OPC-N3 used a total flow rate of 5.5 l min$^{-1}$, a sample flow rate of about 0.28

l min$^{-1}$ and a measurement interval of 1 s. Preliminary experiments indicated that the particles with a nominal diameter of 0.51 $\mu$m were detected in channels 6 to 8 of the GRIMM R11, which cover particle diameters from 0.45 $\mu$m to 0.65 $\mu$m. This corresponds to the second channel of the Alphasense N3 with a nominal diameter range from 0.46 $\mu$m to 0.66 $\mu$m. Air was sampled from the wind tunnel through copper tubing with a circular bent towards the horizontal wind direction and directly connected to the inlets of the particle counters. The inner diameter of the tubing used for the GRIMM OPC was 3 mm with a

total length of 150 mm while the inner diameter for the Alphasense OPC was 5 mm with a total length of 180 mm. Particle concentrations measured by the Alphasense OPCs are in good agreement with those by the GRIMM OPC data (see Figure S1). With a slope of 1.03 and a coefficient of determination $R^2 = 0.91$, the linear regression model indicates a small bias. We conclude that the Alphasense OPC data is physically meaningful despite being a low-cost sensor.

### 2.4    Injection height and flow conditions

Experiments covered four MP injection heights 27 mm, 57 mm, and 255 mm combined with four flow conditions. We define the flow conditions by the voltage supplied to the wind tunnel fans. A minimum voltage of 20 V was necessary for steady rotation of the fans and was defined as flow condition A (FC-A). Flow conditions B, C and D were defined by voltages of 30, 40 and 60 V (FC-B, FC-C, FC-D). The setup resulted in about ten minute data per sampling height for the three Alphasense OPCs, and about three minutes at each of the three sampling heights for the GRIMM OPC, as it cycled through all three

sampling heights. During all experiments, live data of particle concentration was displayed to identify potential problems with particle injection. Some runs had to be aborted when concentrations fell. We suspect the nebulizer produced a few big droplets that clogged the tubing, as after an extended period of operation a subtle wet trail was visible on the roughness elements in front of it. This occurred twice before finishing a measurement cycle, and residual MP was cleaned off before the restart.

### 2.5    Particle flux estimation

The vertical concentration profile, derived from particle concentrations at different heights, allows for estimating the vertical particle flux using flux-gradient similarity theory. This estimation is only valid within the surface layer, where turbulent fluxes are constant with height. For example, the turbulent momentum flux and the wind speed gradient are related as,



$$F_m = -K_m \frac{\partial U}{\partial z} \tag{1}$$

with $F_m$, momentum flux [m$^2$ s$^{-2}$], $K_m$, eddy diffusivity of the momentum flux [m$^2$ s$^{-1}$], U, the mean horizontal wind speed [m s$^{-1}$], $z$, measurement height [m] (Kaimal and Finnigan, 1994). Assuming neutral stratification, the eddy diffusivity $K_m$ can be parameterized as,

$$K_m = \kappa z u_* \tag{2}$$

with $\kappa$ being the von Karman constant (= 0.4), and the friction velocity ($u_*$). Using Equation 2 and integrating Equation 1 between $z_1$ and $z_2$, we obtain,

$$F_m = -\kappa u_* \frac{U_2 - U_1}{ln(z_2/z_1)} \tag{3}$$

As $F_m = u_*^2$, Equation 3 can be re-written as Equation 4,

$$u_* = \kappa \frac{U_2 - U_1}{ln(z_2/z_1)} \tag{4}$$

The mean wind speed becomes zero at height $z_0$, the so-called momentum roughness length. With $U_1 = 0$ at $z_1 = z_0$, we obtain the logarithmic wind profile:

$$U(z) = \frac{u_*}{\kappa} ln(\frac{z}{z_0}) \tag{5}$$

In analogy to Equation 3, the turbulent particle flux $F_c$ [m$^{-2}$ s$^{-1}$] is

$$F_c = -\kappa u_* \frac{c_2 - c_1}{ln(z_2/z_1)} \tag{6}$$

where $c_z$ is the particle concentration at the respective height $z$. Here, we implicitly assume that the eddy diffusivity of scalar particle transport $K_c$ is equal to the eddy diffusivity of momentum transport $K_m$. Note that scalar eddy diffusivity is expected to be larger than for momentum transport, e.g. up to 1.35 times larger for heat transport (Foken, 2016). Thus, our flux values are lower estimates.

## 3   Results and Discussion

### 3.1   Flow conditions

For the slowest fan settings, wind speeds are vertically uniform with U = 0.39 m s$^{-1}$ (FC-A), representing laminar flow conditions (see Figure 2). Starting with FC-B a boundary-layer profile starts to develop in the wind tunnel. The wind speed




**Figure 2.** The logarithmic height (ln(z)) as function of the mean wind speed (U) for the four flow conditions (FC-A to FC-D). The inverse of the slope times the van Karman constant ($\kappa = 0.4$) yields the friction velocity ($u_*$). Extrapolating the data to U = 0 yields the roughness length ($z_0$).

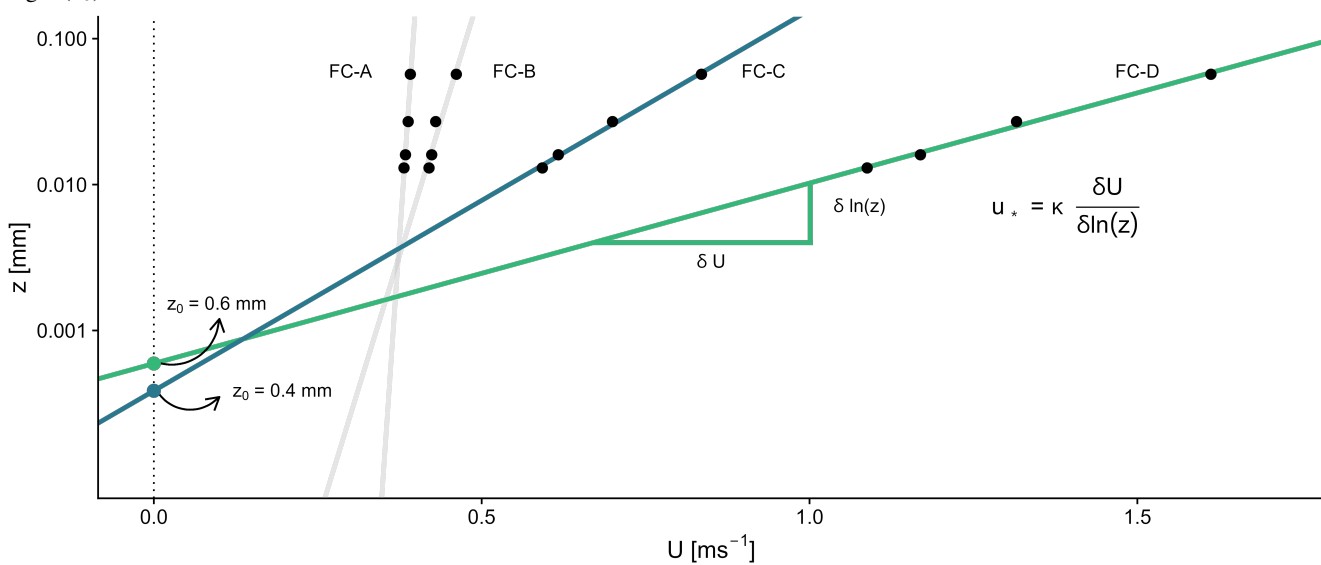

**Table 1.** Friction velocities ($u_*$) and roughness lengths ($z_0$) depending on the flow conditions (FC). Values are calculated according to Equations 4 and 5.

| Flow condition | $u_*$ [m $s^{-1}$] | $z_0$ [mm] |
|---|---|---|
| FC-A | 0 | Not applicable |
| FC-B | 0.01 | Not applicable |
| FC-C | 0.06 | 0.4 |
| FC-D | 0.14 | 0.6 |

gradient increases from FC-B to FC-D. Regressing a linear model to the data, representing ln(z) as a function of mean wind speed U, yields the friction velocity $u_*$ and the roughness length $z_0$, both summarized in Table 1.

From FC-A to FC-D, $u_*$ increases. The roughness length varies from $z_0 = 0.3$ mm to $z_0 = 0.5$ mm, which corresponds with approximately one-twenties of the height of the roughness elements that is about 10 mm. We measured at low wind speeds to observe the effect of laminar flow and turbulent flow on the vertical particle transport.

### 3.2 Vertical particle concentrations

In the control experiments without MP injection, the particle concentration in the observed diameter bins related to the nominal MP diameter of 0.51 $\mu$m ranged from 0.158 to 0.254 cm$^{-3}$ (GRIMM). In contrast, in experiments with MP injection, the particle concentration ranged from 0.49 to 166 cm$^{-3}$, considering all measurements at all different heights. Thus, the baseline

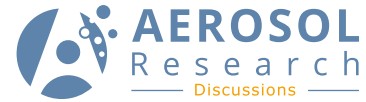

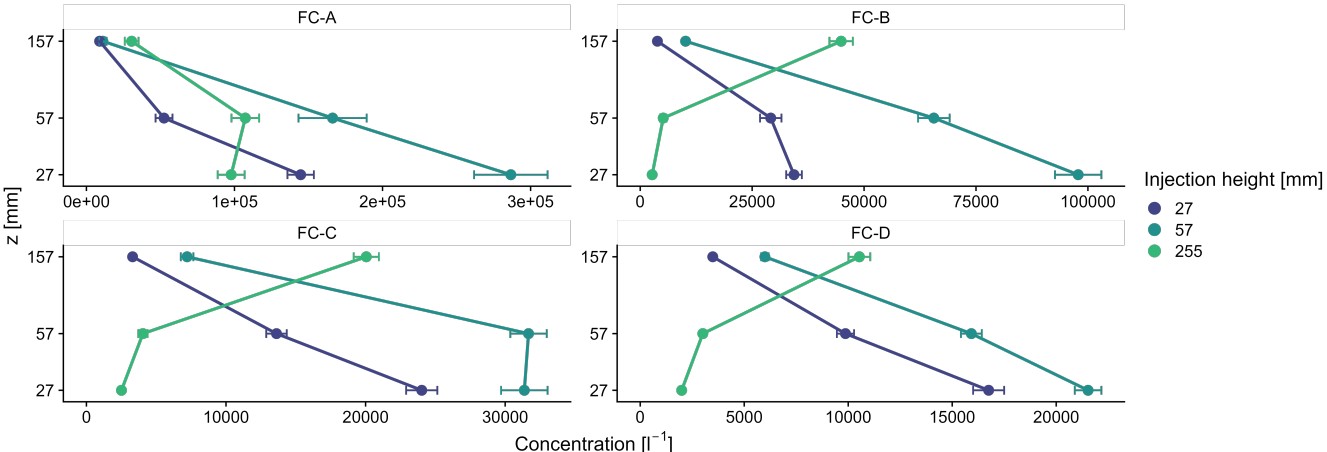

**Figure 3.** Particle concentrations as a function of height for four flow conditions (FC-A to FC-D). The friction velocity increases from FC-A to FC-D. In the individual facets, data is colored corresponding to the injection height. Error bars represent the standard error.

particle concentrations observed in the control experiments are small compared to the particle concentrations with MP injection, and can be neglected. Note that particle concentrations smaller than the baseline concentrations cannot be resolved in our experiments.

Particle concentration profiles for each flow condition and three MP injection heights are shown in Figure 3. In general, particle concentrations are higher at low wind speeds compared to higher wind speeds due to lower dilution. Injection height also influences concentrations, with injection at 27 mm exhibiting overall smaller particle concentrations than the 57 mm height. Unexpectedly, concentrations are highest at 57 mm when injected at 255 mm (FC-A), and highest at 27 mm when injected at 57 mm, challenging previous assumptions. The nebulizer emits water droplets carrying the MP. While the MP is carried by the water droplet, they experience higher gravitational settling as they are part of the relatively larger water droplets. As the water droplets settle down, the MP are released at the height, at which the majority of water droplets evaporated, not at the height of injection (see Figure 4a). Thus, the gravitational settling of the water droplet shifts the release of MP downwards from 255 mm to $\sim$ 57 mm for laminar flow (FC-A). Further, water droplets depositing at the surface before evaporating, inhibit the release of MP into the airflow. Assuming the settling velocity is independent of injection height, higher injection gives more time for evaporation. Hence, injection at 57 mm gives higher concentrations compared to 27 mm, as less MP deposit. For higher wind speeds, turbulence develops and the water droplets settle less downward before evaporating. Hence, the MP are released higher. Thus, injection at 255 mm shows deposition, for FC-B, FC-C and FC-D (see Figure 4b). Further, with increasing wind speed the concentration at the lowest position are more similar for injection at 27 mm and 57 mm, as the MP deposition is decreased. However, for injection at 57 mm the downward shift is still present for FC-D. The results contradict initial assumptions but are explained by the effects of gravitational settling on the particle distribution.



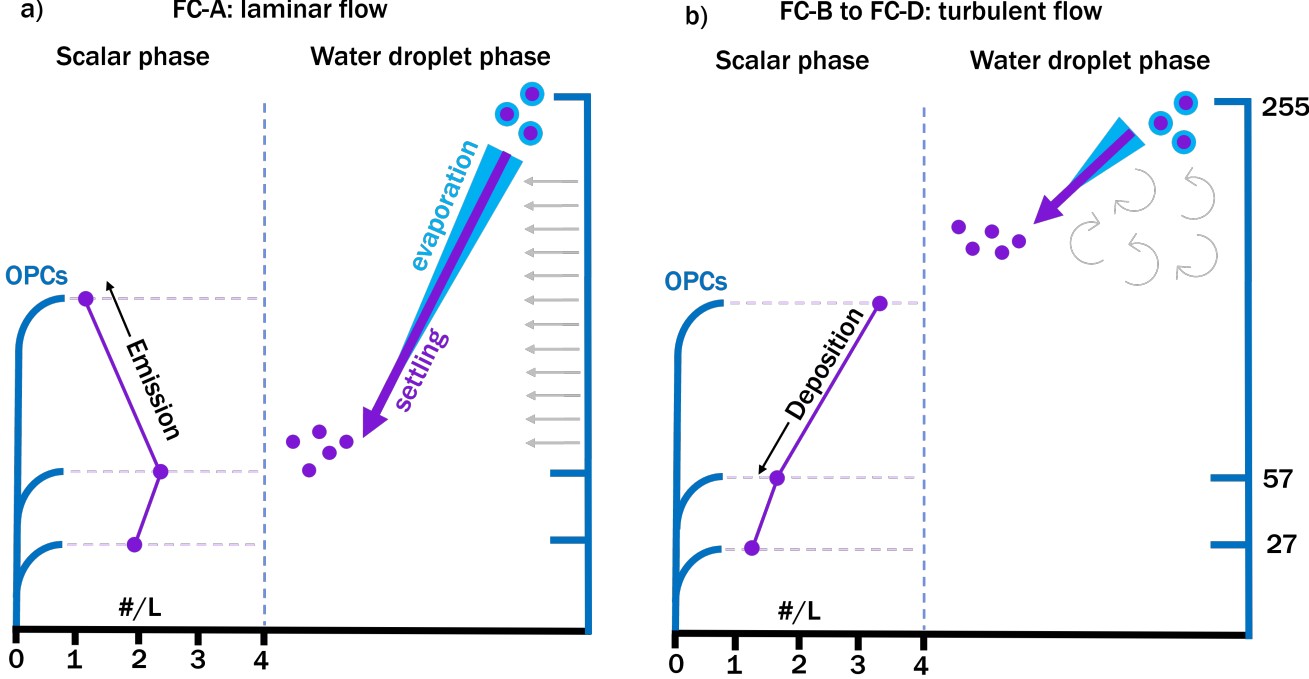

**Figure 4.** Higher settling velocities during the water droplet phase explains the downward shift of the highest particle concentrations. Water droplets carry the polystyrene microspheres (MP). As the water droplets settle down, the MP are released at the height, at which the majority of water droplets evaporated, not at the height of injection. Transitioning to turbulent flow, water droplets settle down less, and the particle concentration profile reverses.

### 3.3 Vertical particle fluxes

Vertical particle fluxes derived from particle concentrations profiles measured by the Alphasense OPCs and the Grimm OPC are similar (see Figure 5). The highest injection at 255 mm results in emission for laminar flows and deposition for turbulent flows. Injection at 57 mm and 27 mm results in emission independent of the flow conditions. Higher wind speeds increase the vertical particle flux for injection at 27 mm. The vertical particle flux for injection at 57 mm and 255 mm changes slightly with increasing wind speed in turbulent conditions. Maximum emission fluxes are observed for an injection height of 57 mm while injection at 27 mm leads to smaller fluxes. For injection at 27 mm, MP are lost to deposition. Hence, the overall particle concentration are smaller, and thus the flux is smaller. With increasing wind speed the deposition decreases, the particle concentration increases and thus the flux increases.

We calculated the deposition velocities for negative particle fluxes, which is the particle flux normalized by its particle concentration. The deposition velocities are similar to other findings for open, flat terrain and they increase with increasing $u_*$ (1.8, 3.74, and 5.73 cm s$^{-1}$ compared to 0.5 - 1 cm s$^{-1}$) (e.g. Sehmel, 1980; Slinn, 1982; Saylor et al., 2019).

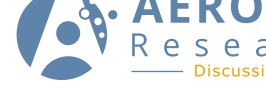

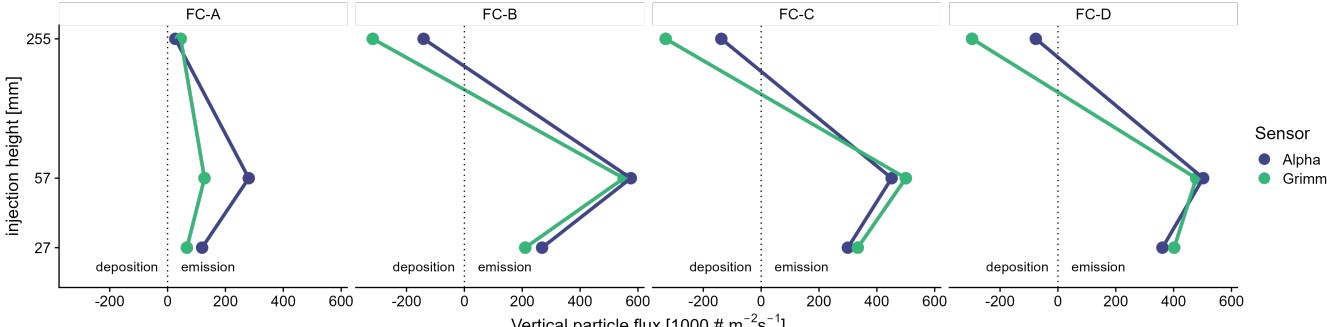

**Figure 5.** The vertical particle flux in relation to the injection height for four flow conditions (FC-A to FC-D). The friction velocity increases from FC-A to FC-D. Positive particle fluxes mean emission and negative particle fluxes mean deposition.

Hence, the results indicate that microplastics in the accumulation mode are transported vertically similar to other particles with likely higher densities in the accumulation mode. However, factory fresh microspheres do not necessarily represent the majority of microplastics found in the environment. In recent studies, fragments and fibers are the predominant shapes of deposited atmospheric MP (Kernchen et al., 2022; Brahney et al., 2020) and it was shown that fibers have significantly lower
settling velocities (Tatsii et al., 2024). Yet, the aerodynamic behavior of such non-spherical particles is often described with an aerodynamic equivalent diameter. Using spherical MP with a nominal diameter of 0.51 $\mu$m was a successful compromise between sufficiently large particle concentrations and reliable optical detection. The large volume flow rate in the wind tunnel leads to strong dilution of particle number concentrations, while the small sampling volume rate of the OPCs requires a minimum number of particles for robust measurements.

## 175  4   Conclusions

Vertical particle fluxes measured by the low-cost Alphasense OPC are comparable to the GRIMM OPC. These low-cost sensors can provide meaningful results for applications where many OPCs are needed, such as concentration profile measurements. We observed the vertical transport of airborne microplastics in laminar and turbulent flow. In laminar flow, gravitational settling in the water droplet phase shifted the release of the airborne microplastics downward. In turbulent flow, the downward shift
decreased and for high injection, it reversed the vertical concentration gradient, resulting in deposition of MP. Overall, recognizing that microplastics share deposition velocities with mineral particles bridges the gap in understanding their environmental behavior. The result supports the use of existing models to evaluate the transport of microplastics in the accumulation mode. The similar deposition velocities imply, that atmospheric transported microplastics can be found in the same places as mineral particles.



*Code and data availability.*  The data that support the findings of this study are available from the corresponding author, EME, upon reasonable request.

## Appendix A

### A1  Alphasense and Grimm

Particle concentrations measured by the Alphasense OPCs are in good agreement with those by the GRIMM OPC data (see

Figure A1). With a slope of 1.03 and a coefficient of determination $R^2 = 0.91$, the linear regression model indicates a small bias. We conclude that the Alphasense OPC data is physically meaningful despite being a low-cost sensor.

*Author contributions.*  EME planned and conducted the wind tunnel experiments, prepared the instrumental setup and revised the manuscript with suggestion by CKT and AH; CG planned and conducted the wind tunnel experiments, prepared the instrumental setup and wrote the manuscript; WB, CKT and AH supervised the writing and experimental process

*Competing interests.*  At least one of the (co-)authors is a member of the editorial board of Aerosol Research

*Acknowledgements.*  Funded by the Deutsche Forschungsgemeinschaft (DFG, German Research Foundation) – Project Number 391977956 – SFB 1357.






**Figure A1.** Particle concentrations measured by the Alphasene optical particle counter (OPCs) as a function of those measured by GRIMM R-11 OPCs. The dashed line, with a slope of one and intercept of zero, signifies perfect agreement. Proximity to this line indicates better agreement between the two OPCs, revealing consistency in measurements, but not necessarily accuracy to the true values.



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
