# Peer review of "Vertical concentrations gradients and transport of airborne microplastics in wind tunnel experiments."

_Aerosol Research, 2024_

## Referee Comment (RC1)

This manuscript outlines an interesting and well executed wind tunnel experiment examining the transport of airborne microplastic particles (MPP) in vertical profile. The results are expected to be useful in the parameterization existing models of aerosol dispersion for use with microplastic particles, particularly since there has been very little experimental work on atmospheric transport.

As outlined below, I have several comments that will likely require minor editing of the manuscript.

Comments

Section 2.1. Description of the wind tunnel.

1. Line 48. I would suggest that you insert "Suction-type" wind tunnel. I am not sure what the total length of 7.3 encompasses. Is this the length from the honeycomb inlet to the fan chamber, or is it the length of the actual test or working section from the spires to the CTA traverse? If it is total length, then you should consider specifying the length of the test section. Although not to scale, you could add this information to the schematic in Fig. 1 for clarification.
2. Line 53. Please provide more detail in regard to the 'pattern of cones and roughness elements' as these are highly pertinent to the aerodynamic roughness length established for the experiment, as well as the turbulence intensity. Such information would also be needed for experimental replication.

Section 2.2 *Aerosol generation*

3. Line 72. How do you know that the droplets "quickly dried" ? How quickly? Over what distance? Is this an assumption, a model projection or a direct observation of a reduction in droplet diameter with the distance of travel from the nozzle of the nebulizer? What was the initial water droplet diameter, as compared to the 52 micron MPP?
4. What natural processes in the atmosphere are simulated by this experimental method of MPP delivery? Can you add several references to make the linkage more direct (natural versus experimental processes)?

Sections 2.3 and 2.4 *Particle measurement; Injection height and flow conditions*

5. Given that the Alphasense and GRIMM OPCs intercalibrate very well as shown in Fig. A1, why did you elect to mount all six sensors at the same downwind location at the end of your test section. Would it not have been more instructive to separate them to examine the fetch effect and locate one at the site of the nebulizer delivery as a quality control check? Maybe I missed it, but it would be further useful to list the distance between the nebulizer and the OPCs directly on your schematic in Fig. 1.

Section 2.5 *Particle flux estimation*

6. Line 101. Are you describing the constant stress region in the lower 20% of the boundary layer?

Section 3.*1 Flow condition*

7. Please add details concerning the thickness (depth) of the boundary layer flow and the reference velocity in the freestream flow in the core of the tunnel working section.
8. I really appreciate the value of drawing a comparison between laminar and turbulent boundary layer flow in regard to suspension of the MPPs. Nice test design. It would be further useful to quantify the reference turbulence level by including in Table 1 the turbulence intensity (rms U/mean U) and the velocity components (horizontal and particularly the vertical) for mean total velocity U.

Section 3.2 *Vertical particle concentration*

9. Line. 135-136. The control concentrations are small but not negligible (i.e., 0.49/0.254 ~2). Consider subtracting them from concentrations with MPPs as a correction, rather than ignoring them.
10. Line 153. "initial assumptions" is too vague. Please consider being more specific here.
11. Figure 4 is very useful in supporting the text with a graphic explanation. You might also consider tying this directly back to Equation 6 (i.e., when $c_2 > c_1$ F is negative and deposition dominates and vice versa).
12. Fig. 5 Do you have a possible explanation for why the Alpha OPC value is consistently greater than the value from the GRIMM at the highest elevation?
13. Define "accumulation mode" which is used here and elsewhere.
14. Line 165. Please provide the Reynolds numbers associated with these terminal velocities.
15. Consider adding a plot (terminal velocity versus particle diameter) in which you compare your MPP data with measurement data obtained for mineral particles of varied size and density (i.e., expand on the context). In relation to this I think several statements on p. 9, which emphasize the similarity between the terminal velocities of the MPPs and mineral particles, are overgeneralized and not well substantiated by only three measurements. Maybe tone them down a little .... 'Our results would suggest that ......., but further experimental work over a greater range of MP particle size and shape is needed.' Also the density of the Polystryene beads is not mentioned in the manuscript but it is about 40% of that for silica, so that in the least, the MPPs in this study should have a lower mass and thus terminal velocity.

Typos and style elements

1. Line 83. Bend rather than "bent"?
2. Fig. 2 Consider removing the point symbols for the zo (roughness length) values as this suggests a direct velocity measurement. Intersection of the dotted line at u m s$^{-1}$ and the profile extrapolation is sufficient.
3. Fig. 3 and legend. Please consider replacing labels FC-A etc with values for the actual friction velocity, which will be far more meaningful for the reader. Be consistent with the axis label format and consider using exponent notation throughout. Replace "Facets" with 'plots'. Report n values (number of test replicates) with standard error. Data 'are'....
4. Line 144. Replace "settle down" with 'descend'.
5. Line 151. "Concentration" should be plural.

---

## Author Comment (AC1)

*We thank the reviewers for their time and comments. Your questions and feedback improved the clarity and quality of our manuscript. Please find our responses in the following document. All comments are individually replied to.*

**Referee 1:**

This manuscript outlines an interesting and well executed wind tunnel experiment examining the transport of airborne microplastic particles (MPP) in vertical profile. The results are expected to be useful in the parameterization existing models of aerosol dispersion for use with microplastic particles, particularly since there has been very little experimental work on atmospheric transport. As outlined below, I have several comments that will likely require minor editing of the manuscript.

*Comments*

Section 2.1. Description of the wind tunnel.

1. Line 48. I would suggest that you insert "Suction-type" wind tunnel. I am not sure what the total length of 7.3 encompasses. Is this the length from the honeycomb inlet to the fan chamber, or is it the length of the actual test or working section from the spires to the CTA traverse? If it is total length, then you should consider specifying the length of the test section. Although not to scale, you could add this information to the schematic in Fig. 1 for clarification.

> *We added that it is a suction-type wind tunnel. Further, we added the total length of the wind tunnel and the working section to the figure. See figure 1*

2. Line 53. Please provide more detail in regard to the 'pattern of cones and roughness elements' as these are highly pertinent to the aerodynamic roughness length established for the experiment, as well as the turbulence intensity. Such information would also be needed for experimental replication.

> *We added a table stating the positioning of the cones and their geometry. See line 184*

Section 2.2 Aerosol generation

3. Line 72. How do you know that the droplets "quickly dried" ? How quickly? Over what distance? Is this an assumption, a model projection or a direct observation of a reduction in droplet diameter with the distance of travel from the nozzle of the nebulizer? What was the initial water droplet diameter, as compared to the 52 micron MPP?

*We assume the droplets dry between injection and measurement since we only detected particles with a 0.5 μm diameter. Therefore, we cannot specify the exact distance or time for evaporation. The distance between injection and measurement was 1.8 m. As no particles larger than 0.5 μm were observed, even at a free stream velocity of 2.7 m/s, the droplets must have dried in under 1 second, which is the time needed to reach the OPC after injection.*

4. What natural processes in the atmosphere are simulated by this experimental method of MPP delivery? Can you add several references to make the linkage more direct (natural versus experimental processes)?

*We added a paragraph about similarities to processes found in nature. We discuss similarities to the emission of microplastics by bubble bursting, rain droplets impacting the ocean surface. See lines 178-183*

Sections 2.3 and 2.4 Particle measurement; Injection height and flow conditions

5. Given that the Alphasense and GRIMM OPCs intercalibrate very well as shown in Fig. A1, why did you elect to mount all six sensors at the same downwind location at the end of your test section. Would it not have been more instructive to separate them to examine the fetch effect and locate one at the site of the nebulizer delivery as a quality control check? Maybe I missed it, but it would be further useful to list the distance between the nebulizer and the OPCs directly on your schematic in Fig. 1.

*We used 3 low-cost OPCs and a single GRIM OPC. Thus, we opted to measure the the vertical concentrations profile with both setups close to the end of the test section. We added the distance between the nebulizer and the OPCs to the wind tunnel schematic. See Figure 1*

Section 2.5 Particle flux estimation

6. Line 101. Are you describing the constant stress region in the lower 20% of the boundary layer?

*Yes. We describe the region, where stresses are constant with height and vertical fluxes are independent of height. We added further description to make it clearer. See line 103*

Section 3.1 Flow condition

7. Please add details concerning the thickness (depth) of the boundary layer flow and the reference velocity in the freestream flow in the core of the tunnel working section.

*We added the boundary layer thickness and freestream velocities to table 1.*

8. I really appreciate the value of drawing a comparison between laminar and turbulent boundary layer flow in regard to suspension of the MPPs. Nice test design. It would be further useful to quantify the reference turbulence level by including in Table 1 the turbulence intensity (rms U/mean U) and the velocity components (horizontal and particularly the vertical) for mean total velocity U.

*We added the turbulence intensity values and the mean horizontal velocity for the top of the boundary layer to table 1.*

Section 3.2 Vertical particle concentration

9. Line. 135-136. The control concentrations are small but not negligible (i.e., 0.49/0.254 ~2). Consider subtracting them from concentrations with MPPs as a correction, rather than ignoring them.

*We subtracted the median baseline concentration from observed particle concentration in experiments with MP injection. See figure 3 and line 137.*

10. Line 153. "initial assumptions" is too vague. Please consider being more specific here.

*We clarified that we expect the highest concentration at the height of particle injection. See lines 154-156*

11. Figure 4 is very useful in supporting the text with a graphic explanation. You might also consider tying this directly back to Equation 6 (i.e., when c2 > c1 Fis negative and deposition dominates and vice versa).

*We added the equation to the plot. See figure 4*

12. Fig. 5 Do you have a possible explanation for why the Alpha OPC value is consistently greater than the value from the GRIMM at the highest elevation?

*The Alphasense OPCs report lower concentrations than the GRIMM OPC at low levels. Consequently, the vertical particle fluxes in figure 5 are consistently smaller when using Alphasense data compared to GRIMM data. During injection at 255, concentrations were low at the bottom and high at the top. The Alphasense overestimates lower concentrations, resulting in a smaller gradient compared to the GRIMM. We added this comment to the supplemental information. See lines 209-212*

13. Define "accumulation mode" which is used here and elsewhere.

*We added a definition of the accumulation mode to the manuscript. See lines 37-38*

14. Line 165. Please provide the Reynolds numbers associated with these terminal velocities.

*The Reynolds numbers are $0.5*10^{-3}$, $1*10^{-3}$ and $2*10^{-3}$, respectively. However, we understand deposition velocities rather as a value indicating how efficient a particle species is removed from the air, than the actual terminal speed of the particles. Thus, we did not add these values to the manuscript, as we believe they do not significantly aid in the interpretation of our results.*

15. Consider adding a plot (terminal velocity versus particle diameter) in which you compare your MPP data with measurement data obtained for mineral particles of varied size and density (i.e., expand on the context). In relation to this I think several statements on p. 9, which emphasize the similarity between the terminal velocities of the MPPs and mineral particles, are overgeneralized and not well substantiated by only three measurements. Maybe tone them down a little …. 'Our results would suggest that ……., but further experimental work over a greater range of MP particle size and shape is needed.' Also the density of the Polystryene beads is not mentioned in the manuscript but it is about 40% of that for silica, so that in the least, the MPPs in this study should have a lower mass and thus terminal velocity.

*We toned down the statement. Discussing the effects of density on deposition velocities is beyond the scope of our manuscript. A detailed comparison would require direct experimental analysis of particles with different densities. See lines 198-191*

*Typos and style elements*
1.  Line 83. Bend rather than "bent"?

*We corrected the typo.*

2. Fig. 2 Consider removing the point symbols for the zo (roughness length) values as this suggests a direct velocity measurement. Intersection of the dotted line at u m s -1 and the profile extrapolation is sufficient.

> *We removed the points.*

3. Fig. 3 and legend. Please consider replacing labels FC-A etc with values for the actual friction velocity, which will be far more meaningful for the reader. Be consistent with the axis label format and consider using exponent notation throughout. Replace "Facets" with 'plots'. Report n values (number of test replicates) with standard error. Data 'are'….

> *We replaced the labels with the respective friction velocities, added consistent break labeling, and revised the caption. See figure 3.*

4. Line 144. Replace "settle down" with 'descend'.

> *We changed the wording.*

5. Line 151. "Concentration" should be plural.

> *We changed the wording.*

This manuscript investigates the vertical transport and deposition velocities of microplastics in the atmosphere through wind tunnel experiments, providing important insights into the environmental behavior of microplastics. The study employs a controlled wind tunnel environment and advanced measurement techniques to produce key results on the concentration gradient and deposition velocity of microplastics at different altitudes. The experimental results show that microplastic deposition is influenced by wind speed and turbulence, while having similar deposition velocities to mineral particles. These findings help fill gaps in the understanding of the environmental behavior of microplastics and support the use of existing models to assess the transport of microplastics in the atmosphere. In this manuscript, the experimental data are valuable, the measurement methods are reliable, the article logic is clear, and the references are appropriate. The manuscript deserves publication after the authors take care of the following moderate revision described below.

1. In section 2.2, is there a more accurate quantification of the $3.9 \times 10^9$ particles per ml obtained by the authors after dilution, or is it solely an estimation based on the manufacturer's provided concentration during microplastics purchase? How does the authors control the uncertainty of the microplastic concentration in the suspension?

   *We calculated the resulting concentration by dilution. The suspension was well mixed and hence the mean concentration measured was stationary. The exact mean value was less important than the relation of the mean concentrations for the varied injection heights and measurement heights.*

2. In lines 110 to 113, was there an omission of a negative sign in Equation 4 when it was derived from Equation 3?

   *We corrected the equation; the negative sign was missing.*

3. In line 116, "-2" should be superscripted.

   *We corrected the typo.*

4. In lines 166-167, the authors state that microplastics in the accumulation mode are transported vertically in a similar way to other particles that may be denser in the accumulation mode. Is it possible to make a detailed comparison directly in the manuscript, which could be a more visual figure?

   *Discussing the effects of density on deposition velocities is beyond the scope of our manuscript. A detailed comparison would require direct experimental analysis of particles with different densities.*

5. The conclusion section could further explore the environmental behavior of microplastics in the atmosphere in relation to other environmental factors to deepen the understanding of the mechanisms of microplastic transport and deposition in the atmosphere.

*We added a paragraph to the conclusion section to deepen the understanding of the mechanisms of microplastic transport. See lines 192-197*